# Optical Properties of PbS Quantum Dots deposited on Glass Employing a Supercritical CO$_2$ Fluid Process

**Bruno Ullrich [1,]\* and Joanna Wang [2]**

[1]  Ullrich Photonics LLC, Manistique, MI 49854, USA
[2]  Department of Chemistry, University of Idaho, Moscow, ID 83844, USA; jswang@uidaho.edu
\*  Correspondence: bruno.ullrich@yahoo.com

**Abstract:** We studied the temperature dependence of the emission and absorption of PbS quantum dots deposited on glass by a supercritical CO$_2$ fluid process. The results show that the emission is ruled by different transitions than the absorption, particularly at cryogenic temperatures. We found indications that these observations can be linked to the PbS concentration used to form the films in conjunction with the capability of the supercritical CO$_2$ method to form dense homogeneous films.

**Keywords:** PbS; supercritical fluid deposition; temperature dependent; photoluminescence; absorption

## 1. Introduction

Light emission of quantum dots (QDs) is of considerable interest for future implementations in photonic-based telecommunication technologies, light emitters, and magneto-optical applications [1–3]. Lead sulfide (PbS) QDs are, in particular, prospective candidates for such applications because their intrinsic features permit the realization of strongly confined particles with a considerable emission range, which can be tuned from near-infrared to visible light by simply shrinking the QD diameter from ~15 nm to ~3 nm [4,5]. Despite progress in the field and the application potential, PbS QDs are still a subject of basic research, in particular, the chemistry of preparation (e.g., in glass or polystyrene, or as colloid, see [4–6] and references therein) and stability of the light emission are issues [7], while the temperature dependence of the photoluminescence (PL) is a chapter of its own. Turyanska et al. [8] reported a rather flat (0.1 meV/K) blue shift of the PL peak up to 150 K, while beyond that temperature an energy shift of 0.3 meV/K was observed. In other publications, an almost constant PL energy position with a variation of ≤110 μeV/K over the temperature range ≈10 K to 300 K was reported [9], and Gaponenko et al. [2]—similar to our own work [10,11]—reported a fairly standard semiconductor-like dependence of the PL peak's energy position on temperature, i.e., only subtle alterations below 40–50 K, followed by a fairly linear growth at elevated temperature. Fan's theory [12,13], although originally derived for bulk materials, describes that behavior very well [11].

In this paper, we investigate the emission of PbS QDs, emphasizing in particular the deep temperature (<50 K) PL properties. The thermal shift of PL is compared with absorption measurements.

## 2. Experimental Section

The PbS QDs were synthesized in our laboratory. Detailed information can be found in the supporting information. The QDs were dispersed on a regular glass substrate by a supercritical CO$_2$ (sc-CO$_2$) fluid process [14]. A typical supercritical fluid deposition system consists of a CO$_2$ source (liquid CO$_2$ tank), a syringe pump (Teledyne ISCO model 260D, Lincoln, NE, USA) with a pump controller (Teledyne ISCO Series D, Lincoln, NE, USA), high-pressure stainless-steel cells, and a collection vessel. The home-made apparatus shown in Figure 1 was used for the PbS QD deposition.

A thoroughly pre-mixed solution of 70 μL of PbS QDs (particle size 4.7 ± 0.5 nm, concentration 40 mg/mL) with the addition of 200 μL toluene solvent was loaded into the apparatus (Figure 1a). After that, a mini glass cup was put upside down on top of the apparatus (Figure 1b), and the sample was inserted into the high-pressure stainless-steel chamber (Figure 1c). The QD film deposition procedure was performed as follows: Liquid $CO_2$ at 60 atm was introduced into the high-pressure chamber gradually until 70 atm. Afterwards, the system temperature was raised from room temperature to 40 °C in order to bring the liquid $CO_2$ in the reaction chamber into the supercritical state. During the heating period, the pressure inside the chamber rose to roughly 140 atm. In order to provide consistency for each deposition procedure, the pressure was increased to 160 atm. The high-pressure apparatus was then left at 40 °C and 160 atm for 30 min in order to ensure equilibrium. The particles were deposited by a gas-antisolvent (GAS) mechanism described previously in the literature [15], where an increasing amount of $CO_2$ altered the polarity of the toluene solvent and it became unfavorable for particle stabilization in the colloid, causing particle precipitation from the solution. Finally, the sc-$CO_2$ was vented slowly and depressurized with a flow rate of around 0.2 mL/min ($CO_2$ becomes gas at the ambient pressure), leaving a deposited PbS QD film behind. Afterwards, the sample was unloaded from the system (Figure 1d). The sample's density was approximately 2.21 mg/cm$^2$ and the average scan height (thickness) of the film was 42.85 kÅ (4.285 μm).

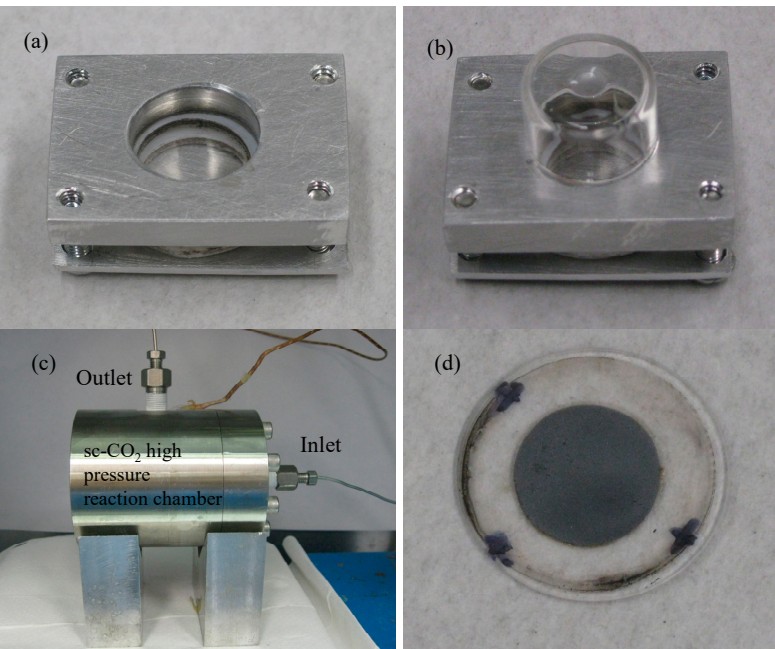

**Figure 1.** Apparatus used to form the samples. (**a**) Substrate holder with a diameter of approximately 1 cm. (**b**) Sample holder with a mini beaker on top. (**c**) sc-$CO_2$ high-pressure reaction chamber, and (**d**) the final product.

On the other hand, films produced by solvent deposition or the drop-casting approach show more irregular and coffee ring-shaped coverage. Comparing solvent and sc-$CO_2$ depositions, the former depends on surface tension effects at the interface of solvent and vapor, while during the sc-$CO_2$ deposition, the supercritical state eliminated the surface tension at the liquid/vapor interface due to surface-wetting instabilities that had detrimental influences on the assembly of low-defect PbS QDs films [14,15]. Consequently, sc-$CO_2$ deposition offers a method for the production of uniform PbS QD films with consistent optical properties. Further information about the sample quality is shown in the Supplementary Material in Figures S1–S3, which show the visible difference of appearance of drop-casted films versus films deposited via the sc-$CO_2$ process, and the x-ray spectrum and the high resolution transmission microscope (HRTEM) image of the film investigated, respectively.

For the PL and absorption measurements, the sample was mounted in an optical closed-cycle helium cryostat with an operating range of 5–300 K. We exposed the sample to the up-converted 532 nm continuous wave (cw) emission of a Nd:YVO4 laser, whereas by adjusting via a lens a laser spot of ~1 mm diameter on the film's surface, the impinging intensity was about 85 W/cm². The PL was collected in reflection geometry and directed into a Fourier transform infrared (FTIR) BOMEM spectrometer equipped with a CaF$_2$ beamsplitter and a nitrogen cooled InSb detector. The film's absorption was measured in transmission geometry by illuminating the sample with a globar light source through a 1 mm aperture, which was closely mounted in front of the surface of the sample. The transmitted signal was collected with a FTIR BOMEM spectrometer as well, in conjunction with a nitrogen cooled Indium Gallium Arsenide (InGaAs) detector and a CaF$_2$ beamsplitter.

## 3. Results and Discussion

Figure 2 shows the PL spectra for the temperature range 5–25 K. The behavior clearly deviated from the semiconductor-like characteristic described in the introductory section above. The rather small temperature increase considerably impacted both the emission intensities and energy positions of the PL peaks. The PL measurements, which were done within a time span of 10–15 min, show clearly visible noise because we kept the excitation level at the noise threshold (signal/noise ≥ 5) in order to exclude optical fatigue as much as possible and to record the intrinsic emission [11]. Figure 3 reveals that the PL peak intensity *I* dropped in proportion to the inverse temperature *T*, i.e., $I = c/T$, where $c = 851.11$ is a constant (goodness of fit 0.997). Excitonic processes frequently cause the inverse temperature dependence of the PL intensity [2]. The PL peak position shifted from 0.78 eV to 0.82 eV, corresponding to a temperature coefficient of 2 meV/K. That energy shift clearly exceeded the bulk band gap shift of 0.5 meV/K [7]. Figure 4 shows the overall temperature dependence of the PL peak. Above 25 K, the energy peak position of the PL peak remained more or less constant until 240–250 K, where it started to increase with the bulk value of 0.5 meV/K.

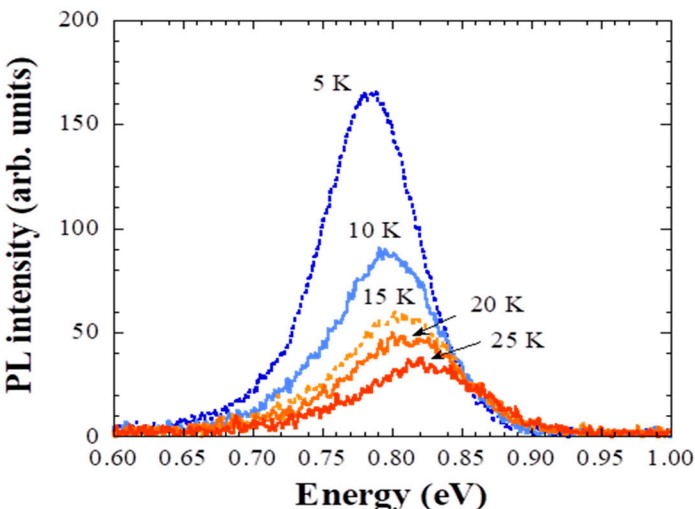

**Figure 2.** PL (photoluminescence) spectra at cryogenic temperatures.

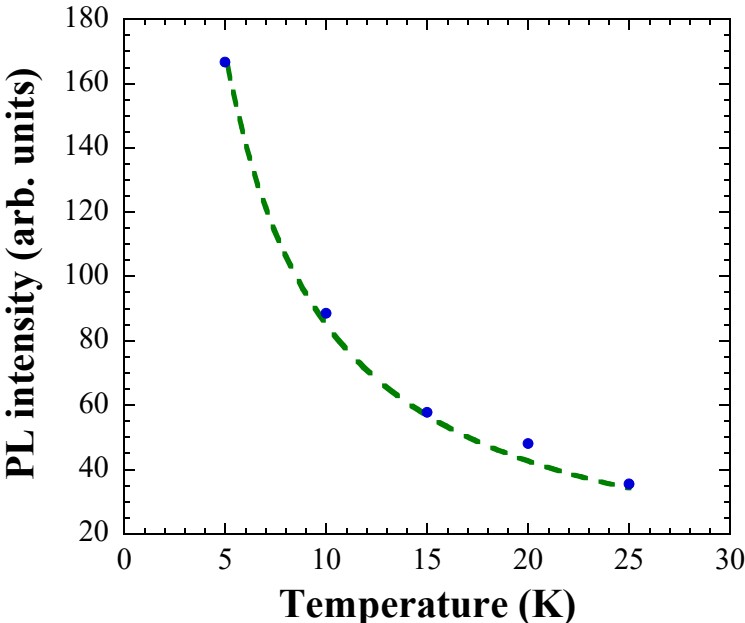

**Figure 3.** PL intensity versus temperature of the spectra in Figure 2. The dashed line is a fit proportional to 1/*T*.

At this point, it is of interest to calculate the band gap energy $E_g(d)$ of free standing QDs of diameter $d = 4.7$ nm. For this purpose, we used the expression introduced by Wang et al. [16,17],

$$E_g(d) = \left[ E_g^2 + \frac{2h^2 E_g}{d^2 m_e^*} \right]^{1/2}, \tag{1}$$

where $E_g$ is the band gap of PbS bulk (0.41 eV at 300 K and 0.28 eV at 77 K) [18], $h$ is Planck's constant, and $m_e^*/m_e = 0.085$ is the effective mass of the electron. For 4.7 nm, the theoretical band gap is 1.22 eV and 0.99 eV at 300 K and 77 K, respectively, i.e., considerably larger than the PL energy peak positions seen in Figures 2 and 4. For comparison, Figure 5 shows the absorbance peak versus temperature. The dashed line is a linear fit of the data with a slope of 0.16 meV/K.

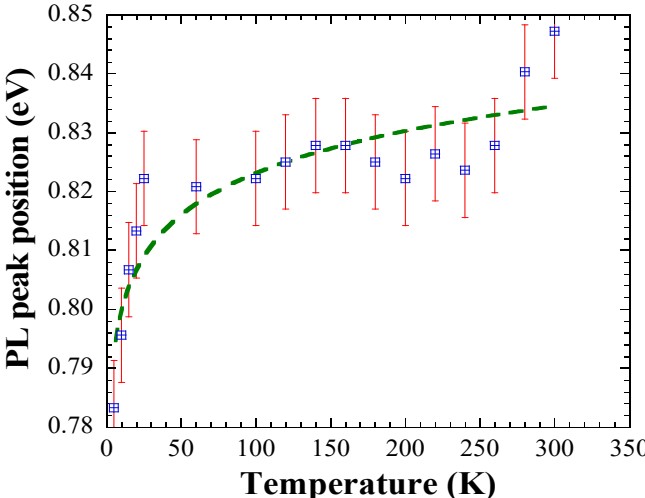

**Figure 4.** PL peak position versus temperature. The dashed line is a guide for the eyes.

In reasonable agreement with the literature [2], at 300 K, the PL peak of ~5 nm particles exhibited a Stokes shift of about 80 meV with respect to the absorbance peak. At 80 K, the absorbance peak

at 0.90 eV in Figure 5 corresponded to a particle size of 5.2 nm, which is within the size tolerance. However, at 300 K the absorbance peak took place at 0.93 eV, corresponding to a particle size of about 6.4 nm. Hence, at cryogenic temperatures the intrinsic features of the QDs rule the absorption, whereas at higher temperatures the band gap of the sample is lowered due to increasing interactions between the QDs. We emphasize here however, that absorption measurements performed on solutions showed the same effect. At room temperature, for solutions with 6.27 mg/mL to 21.95 mg/mL, the absorbance peak stayed constant around 0.93 eV, while for lower concentrations the peak moved gradually towards higher energies and centered around 1.10 eV for 0.78 mg/mL, corresponding again to a particle diameter of 5.2 nm. Hence, in film and solution the increased PbS concentration alters the intrinsic features of the particles. We conclude therefore that the PbS concentration might have a critical cross-over value, which shifts the sample features towards a collective of QDs rather than maintain their separation. The data further stress that the thermal dependence of the PL peak energy is not necessarily linked to the absorption transitions and shows enhanced sensitivity on cryogenic temperature variations. Possibly, the behavior is caused by exciton relaxation processes in combination with carrier trapping, which affects the PL more than the absorption. The carrier trapping might be amplified by the overall modified band structure of closed packed QDs with respect to separated QDs.

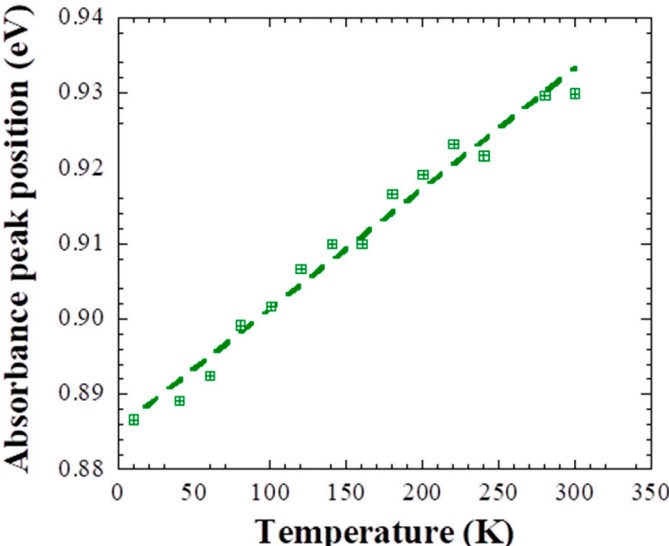

**Figure 5.** Absorbance peak versus temperature. The dashed line is a linear fit of the data.

In our previous works [10,11], regarding the optical properties of ~5 nm QDs on GaAs and glass dispersed by the sc-$CO_2$ procedure as well, the PL peak energy was observed at higher energies, 0.85 eV and 0.79 eV at 5 K, and 0.94 eV and 0.89 eV at 300 K, pointing to more free standing assembled QDs. Indeed, the sample used in the current work possessed a denser and more uniform QD distribution, which is shown by the transmission electron microscope (TEM) image in Figure 6, compared to the QD arrangement of the sample investigated in reference [10]. In order to increase the homogeneity of the QD distribution and to obtain quantitative controllable results during the sc-$CO_2$ process, the following modifications were made: (1) The volume of organic solvent used for the dispersion of nanoparticles was reduced, i.e., not more than 200 μL of toluene was used. (2) The height of the metal cup shown in Figure 1 was increased (the apparatus was made of a thicker aluminum plate). When the sc-$CO_2$ pressure was raised gradually, the PbS precipitated on the bottom before reaching equilibrium of nanoparticles deposited in sc-$CO_2$. (3) The addition of a mini glass cup upside down on the top of the substrate holder (Figure 1b) prevented supercritical fluid vapor deposition from occurring [14]. However, sc-$CO_2$ fluid could still penetrate inside the cup.

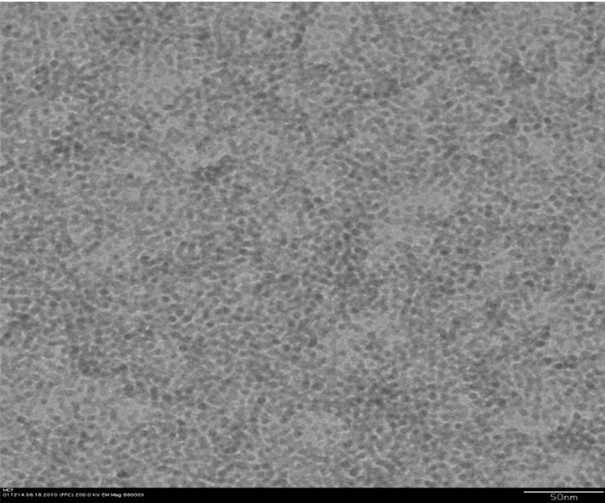

**Figure 6.** TEM (transmission electron microscope) image of the sample investigated. With respect to Figure 1 in reference [10], the PbS QD distribution is more dense and uniform. The bar on the right-hand side in the lower corner indicates the length of 50 nm.

## 4. Conclusions

In summary, we investigated the impact of temperature changes on the emission and absorption properties of PbS QDs deposited on glass with a sc-$CO_2$ fluid process. The results demonstrate that the absorption and PL are not governed by the detailed balance principle, which, at equilibrium, requires equality of the rates of absorption and radiative recombination [19], and only the energy of the band gap transition of the absorption at cryogenic temperatures corresponds to the theoretically expected value. The apparent deviation between the theoretically expected band gap values and the measurements, and in particular, the non-common temperature dependence of the PL maxima, requires more study. The increase in the PbS concentration is the underlying cause of the peculiarities, pointing to a concentration triggered ensemble-like acting of the PbS QDs rather than to features of spatially and electronically separated QDs.

**Supplementary Materials:** The following are available online at http://www.mdpi.com/2076-3417/9/21/4567/s1.

**Author Contributions:** The authors equally contributed to the work.

**Funding:** This research received no external funding.

**Conflicts of Interest:** The authors declare no conflict of interest.

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
