# Peer review of "Optical Properties of PbS Quantum Dots deposited on Glass Employing a Supercritical CO2 Fluid Process"

_applsci, doi:10.3390/app9214567_

Round 1

Reviewer 1 Report

The manuscript contains information on the optical properties of PbS QDs dispersed on a glass substrate made by a supercritical CO2 fluid process. The authors investigate the emission and absorption properties of PbS QDs under different temperature conditions, from cryogenic up to 300 K. The authors conclude, based on the obtained results, that the absorption and PL emission of the QDs rule the theoretical values at cryogenic temperatures, whereas at higher temperatures the band gap of the sample is lowered due to increasing interactions between the QDs.  

In general, the study presents an interesting content, but the discussion about results and some key aspects of the research needs to be further improved. I have some concerns about the discussion on theoretical expected value, and I suggest to revise thoroughly it. The manuscript requires major revision before it is published in this journal. The following issues should be carefully addressed.

Minor comments

Line 38: The authors mention PbS QDs, specifying particle size and dimension. It not clear, however, whether this sample has been bought or synthetized. In the former case, the authors has to specify the name of the company, and the code of the product. If the QDs have been synthetized, the synthesis procedure should be described (eventually in supplementary material) and results of characterization analysis should be reported (XRD, TEM, etc.).

Line 54: Additional details on the employed laser and spectrometer should be inserted. For example, is the laser a mode-locked or q-switched Nd:YVO4? Which is the repetition rate and laser spot size? This information is helpful to the reader to understand the experimental conditions. More details should be added on the measurement set-up: the authors do not specify how absorption and PL are collected (which geometry), and if the showed results are representative of a single or multiple measurements.

Line 66: Typo. At the end of the line there is an extra bracket “This energy shift clearly exceeds the bulk band gap shift of 0.5 meV/K) [6].”

Line 79: Typo. After the word Figure there is an extra dot “considerably larger than the PL energy peak positions seen in Figures. 2 and 4.”

Line 80: It is preferable to use “dashed line” instead of “broken line”.

Line 84: Typo. There is a wrong citation or footnote after word literature “In reasonable agreement with the literature2 at 300 K”.

Major comments

Line 60: The first sentence of Section 3 (“The formed sample showed an unusual temperature dependence of the PL spectra”) anticipates the results illustrated in the following of the paragraph. However, it makes the whole paragraph less explicit. I suggest the authors specify which is the expected usual temperature dependence.

Line 62: The first reported results regard the impact of the temperature variation (between 5 and 25 K) on the emission intensity of the PbS QDs. In particular, it is observed that the PL peak intensity drops in proportion to the inverse temperature. It is important here to specify how the measurements are conducted, because PL from PbS can be subject to photobleaching, if for example, the excited area of the sample is exposed to an energetic beam for a long time. In this case, the reduction of the intensity would be induced, not by a variation of the temperature, but by a quenching of the luminescent emission due to the continuous excitation of the laser. This is a very common effect.

Line 64: The sentence “Excitonic processes frequently cause the inverse temperature dependence of the PL intensity” should be explained or at least sustained with available literature.

Line 74: Wand et al. [12] consider the particles size effects on PbS absorption bands and propose a theoretical model for describing it, extending the electron-hole-in-a-box model that cannot explain the observed size dependence. They obtain the following expression (number 13 in Wang’s paper):

ΔE=[E_g^2+(2(h-bar)^2 E_g π^2)/(d^2 m^* )]^(1/2)

In the expression reported by the authors, there is an error and the second term has to be dived by 2 and not multiply (h-bar is h-Planck constant dived by 2 pi). If the correct formula is used to determine the band gap energy, for particle with dimension of 4.7 nm, it results that Eg(d)=0.7046 eV at 300K. I would therefore highly suggest to the authors to verify the correctness of the expression, of the theoretical results expected for QDs particle, and consequently the discussion related to this.

Line 100: The sentence “The carrier trapping might be amplified by the interaction between the QDs.” should be explained or at least sustained with available literature.

Line 104-116: The last paragraph of the Section 3 points out that, a possible limit in the conducted research, could be represented by the dense QDs distribution and for this reason, the authors modify the process of QDs dispersion in order to increase the homogeneity of distribution and obtain quantitative controllable results. However, I cannot see here the relevance for this work, since any PL or absorbance results are showed on samples obtained with this new method. I suggest either to insert experimental results (if there were some), or improve the discussion related to this last part of the work.

Line 124: In the Conclusion, the authors state that “The results demonstrate that the absorption and PL is not necessarily linked by a basic rule”. However, using the general term “basic rule” is not explanatory itself, and in the same way the “balance principle” is mentioned nowhere in the manuscript. Please be more specific here, referring to explicit physics rule or theoretical study, without being so general.

Author Response

Reviewer 1
Comments and Suggestions for Authors
The manuscript contains information on the optical properties of PbS QDs dispersed on a glass substrate made by a supercritical CO2 fluid process. The authors investigate the emission and absorption properties of PbS QDs under different temperature conditions, from cryogenic up to 300 K. The authors conclude, based on the obtained results, that the absorption and PL emission of the QDs rule the theoretical values at cryogenic temperatures, whereas at higher temperatures the band gap of the sample is lowered due to increasing interactions between the QDs.
In general, the study presents an interesting content, but the discussion about results and some key aspects of the research needs to be further improved. I have some concerns about the discussion on theoretical expected value, and I suggest to revise thoroughly it. The manuscript requires major revision before it is published in this journal. The following issues should be carefully addressed.

Minor comments
Line 38: The authors mention PbS QDs, specifying particle size and dimension. It not clear, however, whether this sample has been bought or synthetized. In the former case, the authors has to specify the name of the company, and the code of the product. If the QDs have been synthetized, the synthesis procedure should be described (eventually in supplementary material) and results of characterization analysis should be reported (XRD, TEM, etc.).

We added supporting information and the sentence “The PbS QDs were synthesized in our laboratory”, and provided more information about the QD preparation (changes are in red).

Line 54: Additional details on the employed laser and spectrometer should be inserted. For example, is the laser a mode-locked or q-switched Nd:YVO4? Which is the repetition rate and laser spot size? This information is helpful to the reader to understand the experimental conditions. More details should be added on the measurement set-up: the authors do not specify how absorption and PL are collected (which geometry), and if the showed results are representative of a single or multiple measurements.

The section is rewritten addressing the concerns of the referee.

Line 66: Typo. At the end of the line there is an extra bracket “This energy shift clearly exceeds the bulk band gap shift of 0.5 meV/K) [6].”

We removed the typo.

Line 79: Typo. After the word Figure there is an extra dot “considerably larger than the PL energy peak positions seen in Figures. 2 and 4.”

We removed the typo.

Line 80: It is preferable to use “dashed line” instead of “broken line”.

We used “dashed line” in the revised manuscript.

Line 84: Typo. There is a wrong citation or footnote after word literature “In reasonable agreement with the literature2 at 300 K”.

We corrected the typo.

Major comments
Line 60: The first sentence of Section 3 (“The formed sample showed an unusual temperature dependence of the PL spectra”) anticipates the results illustrated in the following of the paragraph. However, it makes the whole paragraph less explicit. I suggest the authors specify which is the expected usual temperature dependence.

It is done now in the introduction.

Line 62: The first reported results regard the impact of the temperature variation (between 5 and 25 K) on the emission intensity of the PbS QDs. In particular, it is observed that the PL peak intensity drops in proportion to the inverse temperature. It is important here to specify how the measurements are conducted, because PL from PbS can be subject to photobleaching, if for example, the excited area of the sample is exposed to an energetic beam for a long time. In this case, the reduction of the intensity would be induced, not by a variation of the temperature, but by a quenching of the luminescent emission due to the continuous excitation of the laser. This is a very common effect.

We explained in the revised manuscript why we do not believe that photobleaching plays a major role. Also, the PL peak shifts, it is not only an intensity drop.

Line 64: The sentence “Excitonic processes frequently cause the inverse temperature dependence of the PL intensity” should be explained or at least sustained with available literature.

We referred to reference [2].

Line 74: Wand et al. [12] consider the particles size effects on PbS absorption bands and propose a theoretical model for describing it, extending the electron-hole-in-a-box model that cannot explain the observed size dependence. They obtain the following expression (number 13 in Wang’s paper):
ΔE=[E_g^2+(2(h-bar)^2 E_g π^2)/(d^2 m^* )]^(1/2)

In the expression reported by the authors, there is an error and the second term has to be dived by 2 and not multiply (h-bar is h-Planck constant dived by 2 pi). If the correct formula is used to determine the band gap energy, for particle with dimension of 4.7 nm, it results that Eg(d)=0.7046 eV at 300K. I would therefore highly suggest to the authors to verify the correctness of the expression, of the theoretical results expected for QDs particle, and consequently the discussion related to this.

Our formula is correct. Because Wang wrote it for Eg(R), where R is the radius of the QD. Setting R=d/2 leads to the factor 2 in our expression.

Line 100: The sentence “The carrier trapping might be amplified by the interaction between the QDs.” should be explained or at least sustained with available literature.

We revealed our thought behind the sentence now.

Line 104-116: The last paragraph of the Section 3 points out that, a possible limit in the conducted research, could be represented by the dense QDs distribution and for this reason, the authors modify the process of QDs dispersion in order to increase the homogeneity of distribution and obtain quantitative controllable results. However, I cannot see here the relevance for this work, since any PL or absorbance results are showed on samples obtained with this new method. I suggest either to insert experimental results (if there were some), or improve the discussion related to this last part of the work.

We emphasize now that in previous works the samples did not exhibit the uniformity presented now. See for example Figure 1 in [10] vs. Figure [6] in the current manuscript.

Line 124: In the Conclusion, the authors state that “The results demonstrate that the absorption and PL is not necessarily linked by a basic rule”. However, using the general term “basic rule” is not explanatory itself, and in the same way the “balance principle” is mentioned nowhere in the manuscript. Please be more specific here, referring to explicit physics rule or theoretical study, without being so general.

We clarified the conclusion and provide a reference.

Reviewer 2 Report

The authors show an interesting CO2 supercritical approach on the preparation of PbS QDs films. However, some issues should be addressed before considering it for publication:

(1) in the introduction is missing the comment on the size-dependent optical properties characteristic of QDs and the preparation methods regarding to the possible applications.

(2) along the text "QDs were dispersed on glass substrate" is claimed. However, according with the picture in Figure 1, it seems that they prepared a dispersion in toluene, but finally they work with a PbS QD film on a substrate. please, properly address.

(3) the authors claim that they use glass, but considering that it is a spectroscopic study and the use of low temperatures, I understand that it means quartz or sapphire rather than simple glass. Please, explain.

(4) what is the advantage of using the CO2 supercritical approach against just a simple drop casting method? how compare the results with a just drop cast sample? and therefore it is any change on the PbS QDs nature that the CO2 supercritical approach promotes since they claim no changes in the size according with the PL data at 300K do not match with the initial particles size and the differences observed using different PbS QD solution concentration?

(5) data such as SEM images or thickness of the film are missing. Please provide.

(6) details on the PbS QDs synthesis, organic ligands (or if commercial) are not included in the manuscript. However they are important also to explain the optical behavior of the particles. Authors should detail this.

(7) what are the advantages of the use of the CO2 supercritical approach for preparing QD films in terms of a possible application?

Author Response

(1) in the introduction is missing the comment on the size-dependent optical properties characteristic of QDs and the preparation methods regarding to the possible applications.

We did that in the revised manuscript (changes are in red).

(2) along the text "QDs were dispersed on glass substrate" is claimed. However, according with the picture in Figure 1, it seems that they prepared a dispersion in toluene, but finally they work with a PbS QD film on a substrate. please, properly address.

We changed the sentence “The QD dispersion procedure was done in the following way.” into “The QD film deposition procedure was done as follows.”

We added sentences of “The particles are deposited by a gas-antisolvent (GAS) mechanism described previously in the literature where an increasing amount of CO2 alters the polarity of the toluene solvent and becomes unfavorable for particle stabilization in the colloid, that results in the particles precipitating from solution.” We also changed the sentence “Finally, the outlet was slightly open and depressurized with a flow rate of about 0.2-0.4 mL/min and the sample was unloaded from the system [Figure 1 (d)].” into “Finally, the sc-CO2 was vented slowly and depressurized with a flow rate of around 0.2 mL/min (CO2 becomes gas), leaving deposited PbS QD film behind. Afterwards, that the sample was unloaded from the system [Figure 1 (d)].

(3) the authors claim that they use glass, but considering that it is a spectroscopic study and the use of low temperatures, I understand that it means quartz or sapphire rather than simple glass. Please, explain.

It was in a regular glass. It worked under cryogenic and room temperature conditions.

(4) what is the advantage of using the CO2 supercritical approach against just a simple drop casting method? how compare the results with a just drop cast sample? and therefore it is any change on the PbS QDs nature that the CO2 supercritical approach promotes since they claim no changes in the size according with the PL data at 300K do not match with the initial particles size and the differences observed using different PbS QD solution concentration?

We put the following paragraph into the paper.

On the other hand, films produced by solvent deposition or drop casting approach show irregular and coffee ring-shaped coverage. Comparing solvent and sc-CO2 depositions, the former depends on surface tension effects at the interface of solvent and vapor, while, in sc-CO2 deposition, the supercritical state eliminates the surface tension at the liquid/vapor interface due to surface-wetting instabilities that have detrimental influences on the assembly of low-defect PbS QDs films. Consequently, sc-CO2 deposition offers a method for the production of uniform PbS QD films with consistent optical properties.

We also provide more explanations and pictures in Section 2. Film Formation by sc-CO2
in the Supporting Information.

(5) data such as SEM images or thickness of the film are missing. Please provide.

The film thickness and the density of the film are provided. In order to do SEM measurements, a conductive substrate is required. Because the PbS QDs were directly deposited on glass, there was no SEM measurement easily possible.

(6) details on the PbS QDs synthesis, organic ligands (or if commercial) are not
included in the manuscript. However they are important also to explain the optical behavior of the particles. Authors should detail this.

We added the explanation in the Supporting Information, Section 1. Synthesis Procedure in the supporting Information.

(7) what are the advantages of the use of the CO2 supercritical approach for preparing QD films in terms of a possible application?

We added the explanation in the text (the paragraph next to Figure1) and in the Supporting Information, Section 2. Film Formation.

Reviewer 3 Report

The authors report the optical properties of PbS quantum dots dispersed on Glass employing a supercritical CO2 fluid process. As the optical properties PbS QDs have been investigated thoroughly, I do not see any new results from the PbS QDs itself, the only new is the disperse of QDs on glass, what’s the advantage of this technique? Why not direct use the solution or simple spin-coating or dip-coating on glass slide? As authors stated in the conclusion, the apparent deviation between the theoretically expected band gap values and the measurements, and in particular, the non-common temperature dependence of the PL  maxima, requires more studies. And the ensembly of QDs behavior is just speculation without any direct support (such as HR-TEM). In general, the paper was written as an experimental report which provides not solid evidence to explain the observation, so I do not recommend publishing as its present form.

Author Response

The authors report the optical properties of PbS quantum dots dispersed on Glass employing a supercritical CO2 fluid process. As the optical properties PbS QDs have been investigated thoroughly, I do not see any new results from the PbS QDs itself, the only new is the disperse of QDs on glass, what’s the advantage of this technique? Why not direct use the solution or simple spin-coating or dip-coating on glass slide? As authors stated in the conclusion, the apparent deviation between the theoretically expected band gap values and the measurements, and in particular, the non-common temperature dependence of the PL maxima, requires more studies. And the ensembly of QDs behavior is just speculation without any direct support (such as HR-TEM). In general, the paper was written as an experimental report which provides not solid evidence to explain the observation, so I do not recommend publishing as its present form.

We added the explanation in the text and in the Supporting Information in Section 2. Film Formation. 

We have discussed the results of spin coating, dip coating, drop-casting vs. sc-CO2 approaches and provide evidence to support our conclusion.

Figure S-3 in the Supporting Information shows HRTEM of PbS QDs.

Round 2

Reviewer 1 Report

The authors have properly addressed all the required changes and comments.

Reviewer 2 Report

Authors have addressed the issues asked.